# Selective Deferred Routing: Enabling Cost-Efficient Collaboration between Local SLMs and Remote LLMs

## Abstract

The rapid advancement of large language models (LLMs) has led to remarkable performance across diverse domains such as question answering, creative writing, programming, etc., making them indispensable assistants in daily life and work. Currently, LLM services are primarily accessed in two ways: (i) paid access to cloud-hosted LLMs, which are powerful but introduce nontrivial cost; and (ii) deployment of small language models (SLMs) on personal devices or small clusters, which, while less powerful, are sufficient for handling relatively simple tasks. To achieve a balanced trade-off between monetary cost and task performance, we propose Selective Deferred Routing, a paradigm that enables cost-efficient collaboration between local SLMs and remote LLMs. In this framework, a user request is first processed by the local SLM, which not only generates a preliminary response but also provides rich semantic representations of the request. A lightweight decider module then leverages this information to either adopt the initial response or route the request in a single step to the most suitable remote LLM for a higher-quality response. Extensive experiments on 5 LLMs and 3 datasets demonstrate that our approach consistently outperforms existing multi-LLM collaboration methods across different cost–performance trade-off preferences.

## 1 Introduction

Recent advancements in pre-trained language models based on the Transformer architecture (Vaswani et al., 2017) have demonstrated remarkable capabilities across a wide range of tasks, including question answering, creative writing, programming, and even complex reasoning (Brown et al., 2020). Consequently, Large Language Models (LLMs) are increasingly used as indispensable assistants in both everyday and professional settings.

Nowadays, most end-users obtain LLM services via two approaches. The first is paid access to cloud-hosted LLMs: very large models running on massive remote compute infrastructure that offer the strongest capabilities but incur nontrivial monetary cost. The second is local deployment: running relatively smaller LLMs on personal devices or on small on-premises clusters, enabling essentially cost-free access to inference at the point of use. Although smaller models are typically less capable than large cloud models, they are often sufficient for routine tasks that do not require extensive reasoning or access to very large knowledge bases (Subramanian et al., 2025). And thanks to a set of techniques like low-precision quantization (Dettmers et al., 2022; 2024), knowledge distillation (Hinton et al., 2015; Sanh et al., 2019; Gu et al., 2023), etc., the feasibility and utility of local deployment have improved substantially.

On the other hand, user demands for LLM services are highly diverse and span different levels of complexity, with a substantial portion concerning everyday advice and assistance (Chatterji et al., 2025). Consequently, dynamically choosing or combining the above two LLMs using approaches according to the task characteristics offers a promising way to trade off monetary cost and task performance. For clarity and brevity in the remainder of this paper, we refer to paid, cloud-hosted LLMs as *remote LLMs*, and to smaller LLMs hosted on personal devices or small on-premises clusters as *local SLMs*. Moreover, considering the computational resource limitations of individuals and small institutions, we assume that there is always one local SLM in the system.

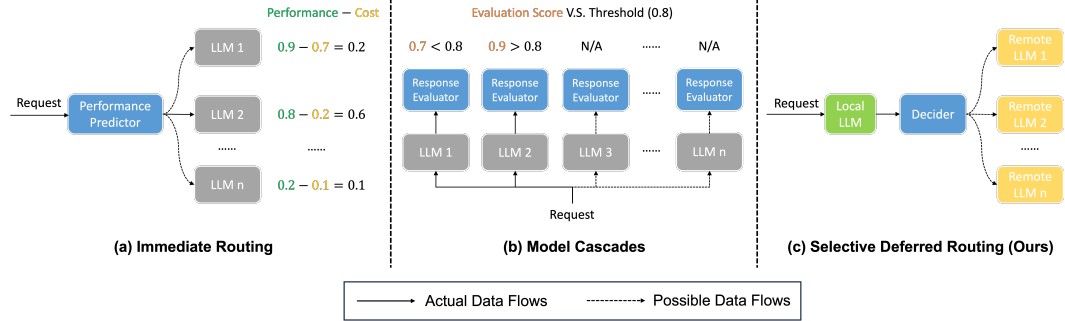

Figure 1: The workflow overview of two existing multi-LLM collaboration paradigms (Immediate Routing and Model Cascades) and the proposed Selective Deferred Routing.

Focusing on cost-efficient multi-LLM collaboration with respect to task characteristics, existing studies have primarily explored two paradigms. The first is *Immediate Routing* (Ong et al., 2024; Ding et al., 2024; Hu et al., 2024; Ding et al., 2025; Song et al., 2025). As shown in Figure 1 (a), a trained performance prediction model takes the user's prompt as input to forecast the performance of each candidate LLM. Then, a term measuring the estimated LLM inference cost will be subtracted from the predicted performance. The model with the highest resulting score is selected to handle the user request. The workflow is straightforward and intuitive; however, because it relies solely on the user prompt and lacks additional decision-making cues, its performance in practice is often limited.

The second paradigm is *Model Cascades* (Chen et al., 2023; Aggarwal et al., 2023; Yue et al., 2023; Nie et al., 2024; Zhang et al., 2024b). As shown in Figure 1 (b), it organizes all available LLMs into a sequential chain, with the assumption that LLMs positioned later in the chain possess greater capabilities but also incur higher costs. The system sequentially invokes LLMs to handle the user request, then evaluates their response by a trained model. If any evaluation result exceeds a predefined threshold, the response will be returned early to the user without invoking the remaining LLMs. This paradigm stems from the speculative notion that low-cost models may suffice to handle a given task. However, its linear structure can lead to the invocation of all available LLMs in the worst case, resulting in substantial costs and increased response latency. Moreover, the assumption of a strict price–performance ordering among LLMs does not hold in today's heterogeneous model market.

It is also noteworthy that both of the paradigms discussed above treat all available LLMs as black boxes. In the practical scenarios considered in this paper, however, local SLMs can provide information beyond textual responses (e.g. hidden states), which can be leveraged to inform later decisions. Another line of research has focused on the characteristics of local SLMs and remote LLMs and their collaboration (He et al., 2024; Zhang et al., 2024a; Hao et al., 2024; Liu et al., 2025; Xu et al., 2024), but primarily emphasizes system-level metrics such as latency, throughput, and load balancing, rather than the trade-off between monetary cost and task performance.

Based on the above analysis and observations, we propose Selective Deferred Routing (SDR), a paradigm that enables cost-efficient collaboration between local SLMs and remote LLMs. As shown in Figure 1 (c), it initially delegates user requests to the local SLM, which not only produces a textual response but also provides rich feature representations of the task. A lightweight decider module then utilizes the enriched information to make a selective routing decision, in which those already satisfactory responses from the local SLM will be directly returned, while the others will be routed to the most suitable remote LLM for a higher-quality response.

To obtain a sufficiently powerful decider, we begin with a simplified yet fundamental formulation, the Local-Remote Collaboration problem, that involves only one remote LLM. In this setting, the selective routing problem can be essentially formulated as an AUC (Area Under Curve) optimization problem, which yields a more effective training objective than naive quality prediction. Moreover, we design the decider module by reusing the architecture of a single transformer block from the local LLM. This allows us to directly initialize it with pretrained parameters from the local SLM, thereby enhancing its ability to exploit the information from SLM's hidden states. Finally, this design can be efficiently extended to scenarios with multiple remote LLMs. In such cases, we adopt an aggregation strategy among multiple deciders to determine whether and how to select among the available remote LLMs.

The main contributions of this work are summarized below:

- We propose Selective Deferred Routing (SDR), a novel collaboration paradigm between local SLMs and remote LLMs, which can achieve a more effective balance between cost and performance.

- We build a strong decider module by revealing that the Local-Remote Collaboration problem can be formulated as an AUC optimization problem, which leads to a more effective training objective. And the design can be further extended to scenarios with multiple remote LLMs.

- Extensive experiments conducted on 5 LLMs and 3 datasets demonstrate that SDR consistently outperforms existing multi-LLM collaboration methods across different cost–performance trade-off preferences.

## 2 RELATED WORK

**Cost-Efficient Multi-LLM Collaboration.** A substantial body of recent work has demonstrated interest in collaborative systems involving multiple LLMs with varying costs and performance characteristics. As shown in Figure 1, Immediate Routing relies solely on the user prompt to predict the performance of all available LLMs. Hybrid LLM (Ding et al., 2024) fine-tunes a BERT-based model to predict the performance of two available LLMs for a given user request. RouteLLM (Ong et al., 2024) also focuses on scenarios with two available LLMs and leverages human preference labels between them from Chatbot Arena (Chiang et al., 2024). It evaluates three training strategies, including similarity-weighted ranking, matrix factorization, and a BERT classifier, to learn effective routing between the larger and smaller LLMs. RouterBench (Hu et al., 2024) employs Sentence Transformers (Reimers & Gurevych, 2019) to obtain embeddings of the user prompt, followed by the application of KNN and MLP models to map these embeddings to all available LLMs' performance. Another collaboration paradigm, Model Cascades, organizes all available LLMs sequentially, then keeps invoking the next model in the sequence until a sufficiently satisfactory response is obtained. FrugalGPT (Chen et al., 2023) fine-tunes a BERT-based model to evaluate the quality of LLM responses. AutoMix (Aggarwal et al., 2023) attempts to have LLMs provide a confidence score alongside their answers, which serves as a proxy for response quality. Mixture of Thought (Yue et al., 2023) leverages multiple sampling: for each user request, multiple responses are sampled from a single LLM, and the consistency among these responses is used to estimate the model's confidence, which in turn acts as a proxy for response quality. In addition, some work focuses on more specialized scenarios, such as online learning on streaming data (Nie et al., 2024), reinforcement learning under budget-constrained settings (Zhang et al., 2024b), etc.

As noted in Section 1, these methods treat all available LLMs as black boxes, considering only their textual inputs and outputs. In practical scenarios, however, local SLMs can provide richer information that can be leveraged for decision-making. Moreover, the workflow and organizational structure of these methods do not align well with the characteristics of today's heterogeneous model market or with user requirements.

**System Design between Local SLMs and Remote LLMs.** Another line of research focuses on leveraging the distinct characteristics of local SLMs and global LLMs for system-level design. Computational offloading (He et al., 2024; Zhang et al., 2024a; Hao et al., 2024) aims to schedule tasks across devices based on their computational resources, workloads, and other characteristics, in order to optimize system-level metrics such as load balancing and user-side latency. Edge–cloud collaborative speculative decoding (Liu et al., 2025; Xu et al., 2024) treats local SLM as draft models while performing token validation on the remote LLM, thereby increasing system throughput without compromising performance. In general, although these studies also recognize the unique value of local SLMs in collaborative systems, they primarily focus on optimizing service-level metrics such as latency, throughput, and load balancing, rather than balancing task performance and monetary cost, making them a parallel line of research to our work.

## 3 SELECTIVE DEFERRED ROUTING

In this section, we present the design and implementation details of Selective Deferred Routing. In Section 3.1, we begin with a simplified setting, the Local-Remote Collaboration problem, which involves the local SLM and one remote LLM. It allows for a more principled analysis under a well-structured scenario in Section 3.2. In Section 3.3, we provide the design details of our decider module. In Section 3.4, we extend the design to the scenario with multiple remote LLMs, providing a practical solution that better aligns with real-world user demands.

### 3.1 LOCAL-REMOTE COLLABORATION PROBLEM

We begin with a simplified yet fundamental setting. Consider one local SLM, denoted as $M_S$, and one remote LLM, denoted as $M_L$. We also have a dataset $\mathcal{D} = (\mathcal{Q}, p)$, consisting of a query set $\mathcal{Q}$ and a performance evaluator $p : (\mathcal{Q}, \mathcal{A}) \to [0, 1]$ measuring the quality of an answer to the given query. The implementation of $p$ depends on the dataset. For datasets with binary labeling, we set $p = 1$ for correct answers and $p = 0$ otherwise. For datasets with graded evaluation metrics, we normalize the resulted metrics to the range $[0, 1]$ to ensure consistency across datasets.

Each model $M \in \{M_S, M_L\}$ takes a user query $q \in \mathcal{Q}$ as input and produces an output. For the local SLM, the output consists of a textual answer $a \in \mathcal{A}$, together with a set of intermediate representations $o \in \mathcal{O}$, such as the hidden states of transformer layers. In contrast, the remote LLM only produces the textual answer.

As shown in Figure 1 (c), for each query $q \in \mathcal{Q}$, the local SLM produces an output $M_S(q) = (a, o)$. The decider module then performs a selective routing decision, which in the simplified setting reduces to a binary choice: either accept the local SLM's answer directly or forward the query to the remote LLM. We define this process as *Binary Selective Routing* BSR $: (\mathcal{Q}, \mathcal{A}, \mathcal{O}) \to \{M_S, M_L\}$.

The binary selective routing result can be obtained following the binary classification paradigm, in which we first use a scoring model to generate a score:

$$S : (\mathcal{Q}, \mathcal{A}, \mathcal{O}) \to [0, 1], \tag{1}$$

which represents the confidence of adopting the local SLM's output. The selective routing decision is then made by specifying a threshold $\alpha \in [0, 1]$:

$$\text{BSR}^\alpha(q, a, o) = \begin{cases} M_S, & \text{if } S(q, a, o) \geq \alpha \\ M_L, & \text{otherwise.} \end{cases} \tag{2}$$

In practice, different choices of the threshold $\alpha$ reflect the user's preference for either lower cost or higher performance.

Finally, we can formalize the notions of the overall cost and task performance induced by the workflow defined above. For clarity and practical relevance, we measure the user's cost as the proportion of queries that are routed to the remote LLM:

$$\text{Cost}(\mathcal{Q}, \text{BSR}^\alpha) = \frac{1}{|\mathcal{Q}|} \sum_{q \in \mathcal{Q}} \mathbf{1}\{\text{BSR}^\alpha(q, M_S(q)) = M_L\}. \tag{3}$$

Then we define the task performance as the average output quality of the LLMs to which the queries are routed:

$$\text{Perf}(\mathcal{Q}, \text{BSR}^\alpha) = \frac{1}{|\mathcal{Q}|} \sum_{q \in \mathcal{Q}} p(q, \text{BSR}^\alpha(q, M_S(q))(q)). \tag{4}$$

### 3.2 AUC OPTIMIZATION: METRIC AND METHOD

Based on the above formulation, we need to evaluate the BSR module while applying different $\alpha$. We first consider boundary cases. Specifically, $\text{Perf}(\mathcal{Q}, \text{BSR}^0)$ corresponds to the overall performance of the local SLM, as all queries are not routed to the remote LLM. Similarly, $\text{Perf}(\mathcal{Q}, \text{BSR}^1)$ represents the overall performance of the remote LLM.

For any other $\alpha \in (0, 1)$, the BSR module routes a fraction of queries to the remote LLM, yielding a cost $\text{Cost}(\mathcal{Q}, \text{BSR}^\alpha) \in (0, 1)$ and a performance $\text{Perf}(\mathcal{Q}, \text{BSR}^\alpha)$. We first define the *Performance*

*Gain* (PG) to capture the relative improvement, which is normalized on a relative scale between the local SLM ($\text{Perf}(\mathcal{Q}, \text{BSR}^0)$) and the remote LLM ($\text{Perf}(\mathcal{Q}, \text{BSR}^1)$):

$$\text{PG}(\mathcal{Q}, \text{BSR}^\alpha) = \frac{\text{Perf}(\mathcal{Q}, \text{BSR}^\alpha) - \text{Perf}(\mathcal{Q}, \text{BSR}^0)}{\text{Perf}(\mathcal{Q}, \text{BSR}^1) - \text{Perf}(\mathcal{Q}, \text{BSR}^0)}. \tag{5}$$

In fact, each choice of $\alpha$ can be interpreted as the outcome for a user with budget $\text{Cost}(\mathcal{Q}, \text{BSR}^\alpha)$, yielding a return quantified by $\text{PG}(\mathcal{Q}, \text{BSR}^\alpha)$. Therefore, it's natural and also convenient to denote $c$ as a shorthand for $\text{Cost}(\mathcal{Q}, \text{BSR}^\alpha)$, and write $\text{PG}(c)$ for the corresponding $\text{PG}(\mathcal{Q}, \text{BSR}^\alpha)$. Let $\Pr(c)$ denote the probability density function of user budgets over the normalized cost domain $c \in [0, 1]$. The *expected performance gain* under this budget distribution is then defined as:

$$\mathbb{E}_{c\sim\Pr(c)}[\text{PG}(c)] = \int_0^1 \Pr(c)\text{PG}(c)\mathrm{d}c. \tag{6}$$

In this work, we regard this expected performance gain as the principal metric for evaluating the effectiveness of the BSR module. For simplicity and without loss of representativeness, we assume that user budgets are uniformly distributed over the cost range.

Geometrically, this metric corresponds to the area under the cost–performance trade-off curve (AUC). As shown in Figure 2, varying $\alpha$ traces out points $(c, \text{PG}(c))$. Intuitively, a BSR module achieving a larger AUC (the blue curve) directly indicates that users can attain performance close to that of the remote LLM even under a limited budget.

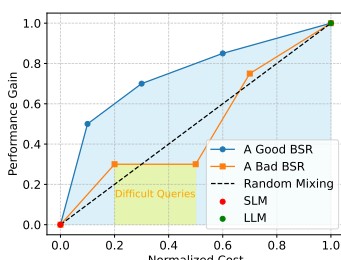

Figure 2: An example Cost-Perf Curve obtained from two possible BSR modules with 5 sampled $\alpha$.

Next, we aim to train a parameterized scoring model to optimize the expected performance gain, i.e., cost-performance AUC. A straightforward training objective is to predict the quality of the local SLM's answer, which is widely adopted in previous work. Answers with sufficiently high predicted quality (above the threshold) are returned directly, while queries with low-quality answers are routed to the remote LLM in the hope of obtaining a higher-quality response. During training, the scoring model can be optimized using a standard cross-entropy loss between its output and the correctness of the SLM's answer.

While this approach is intuitive and easy to implement, it does not effectively optimize the AUC metric in practice. For instance, consider a particularly difficult query for which neither the local SLM nor the remote LLM can generate a high-quality answer. In such cases, the optimal choice is to return the local answer directly to save cost. However, under the aforementioned training paradigm, as shaded in the yellow area of Figure 2, the decider would still be biased toward routing the query to the remote LLM with a relatively low $\alpha$, leading to unnecessary expense without improving performance.

From the above example, we draw a simple but important insight: in binary selective routing, the decisive factor is not the absolute performance of either the SLM or the LLM, but rather their *performance gap*. We define this gap for a given query $q$ as:

$$\text{Gap}(q) = p(q, M_L(q)) - p(q, M_S(q)), \forall q \in \mathcal{Q}. \tag{7}$$

We then state the following result:

**Theorem 1.** *The routing decisions produced by the BSR module yield the optimal cost–performance AUC if and only if, for all pairs of queries $q_i, q_j \in \mathcal{Q}$,*

$$Gap(q_i) \geq Gap(q_j) \implies S(q_i, M_S(q_i)) \leq S(q_j, M_S(q_j)).$$

We defer the detailed proof to Appendix A and instead provide an intuitive explanation here. As the threshold $\alpha$ increases, queries are progressively routed to the remote LLM in ascending order of their scores. Queries with larger performance gaps contribute steeper slopes on the cost–performance curve. Consequently, routing such high-gap queries to the remote LLM earlier, i.e., toward the left side of Figure 2, yields a more convex curve, thereby resulting in a higher AUC.

It follows that optimizing the scoring model from the perspective of ordering consistency is the most direct way to improve the cost–performance AUC. In fact, ranking optimization is a well-studied problem (Chen et al., 2009; Buffoni et al., 2011), which aims to train models whose predicted scores preserve the relative ordering of labels. However, classical optimization approaches typically require large optimization batches to obtain sufficiently representative pairwise comparison information across the dataset. Under our problem setting, the scoring model relies on information derived from the local SLM, making large-batch training on consumer-grade devices impractical. Thus, to retain representativeness while reducing complexity, we reduce the performance gap into a binary space to obtain *Binary Gap* (BG):

$$\text{BG(q)} = \mathbf{1}\{\text{Gap}(q) > \beta\}, \forall q \in \mathcal{Q}. \tag{8}$$

That is, we categorize a query $q$ as one where the remote LLM holds a significant advantage whenever its performance surpasses that of the local SLM by more than the threshold $\beta$; otherwise, it is regarded as having no clear advantage. For this binary characterization of the performance gap, a direct corollary of Theorem 1 can be established:

**Corollary 1.** *The routing decisions produced by the BSR module yield the optimal cost–performance AUC if and only if, for all $q^+ \in \mathcal{Q}$ with $BG(q^+) = 0$ and all $q^- \in \mathcal{Q}$ with $BG(q^-) = 1$, it holds that*

$$S(q^+, M_S(q^+)) > S(q^-, M_S(q^-)).$$

The optimization objective for the scoring model is now fully aligned with the classical goal of maximizing the Area Under the ROC Curve in binary classification tasks. Under this simplified setting, a variety of methods have been proposed that are particularly suitable for optimization for smaller batches (Ying et al., 2016; Yuan et al., 2021; Sharma et al., 2023). We adapt the approach of Yuan et al. (2021), which finally leads to our following training objective:

$$\mathcal{L} = \frac{1}{|B^+|} \sum_{q^+ \in B^+} (S(q^+, M_S(q)) - a)^2 + \frac{1}{|B^-|} \sum_{q^- \in B^-} (S(q^-, M_S(q)) - b)^2 + (m - (a-b))^2_+. \tag{9}$$

Here, $B^+$ denotes the subset of queries within the batch where $\text{BG}(q) = 0$, and $B^-$ denotes those where $\text{BG}(q) = 1$. The parameters $a$ and $b$ are trainable anchors, while $m$ is a predefined margin. The first two terms of the loss encourage the scores of queries in $B^+$ and $B^-$ to align closely with the anchors $a$ and $b$, respectively. The third term enforces a separation between $a$ and $b$ such that their distance is at least $m$. By introducing these two learnable anchors, this training objective stabilizes AUC optimization under small batch sizes and further enables loss aggregation across multiple device batches.

### 3.3 SCORING MODEL DESIGN

We now describe the details of the scoring model, which is carefully designed to fully utilize the rich information produced by the local SLM while remaining lightweight and practical for on-device deployment.

As illustrated in Figure 3, it consists of a single transformer layer followed by an MLP. By adopting a transformer architecture consistent with that of the local SLM, the model can leverage per-token hidden state outputs, effectively capturing the contextual information of the user query and the local SLM's response. Further-

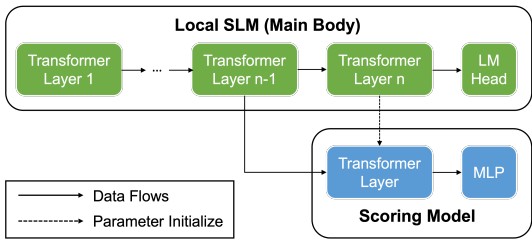

Figure 3: The architecture of the scoring model.

more, we initialize the transformer layer of the scoring model using the pretrained parameters from the last layer of the local SLM, allowing it to better align with the hidden state distribution from the penultimate layer of the local SLM and enhance data efficiency. Overall, this design maximizes the utilization of the local SLM's capability, even including information encoded during pretraining.

Importantly, the scoring model is lightweight and practical for deployment on consumer-grade devices. For instance, when using Qwen-2.5-7B (Qwen Team, 2024) as the local SLM, which comprises 28 transformer layers with approximately 0.233B parameters per layer, the corresponding

scoring model's parameter count is orders of magnitude smaller than that of the full local SLM, leading to a minimal memory footprint. Moreover, its computation can be performed in parallel with the latter part of the local SLM, introducing negligible additional latency.

We also explored scoring models that incorporate multiple transformer layers. However, as detailed in Appendix C.1, increasing the number of layers does not lead to consistent improvements in evaluation. Considering the practical need to keep the module lightweight for efficient deployment, we ultimately adopt the above single-layer design.

### 3.4 Towards Collaboration with Multiple Remote LLMs

To further match the real user demands, we extend the aforementioned design from the simplified setting to the multiple remote LLMs setting. Suppose we now have $N$ remote LLMs, $\{M_L^{(1)}, M_L^{(2)}, \cdots, M_L^{(N)}\}$. Since a single scoring model can measure the relative quality gap between the local SLM and one remote LLM, we can train a corresponding scoring model for each remote LLM: $\{S^{(1)}, S^{(2)}, \cdots, S^{(N)}\}$. Given the lightweight nature of each scoring model, it is practical to run them in parallel during deployment. Accordingly, we define the *Selective Routing* (SR) decision with a given threshold $\alpha$ as follows. First, for each remote LLM $M_L^{(i)}$, we compute its score for a given query $q$ as $S^{(i)}(q, M_S(q))$. We then calculate the average score corresponding to all remote LLMs:

$$\overline{S}(q, M_S(q)) = \frac{1}{N} \sum_{i=1}^{N} S^{(i)}(q, M_S(q)). \tag{10}$$

The *selective routing* (SR) decision is then made by specifying a threshold $\alpha \in [0, 1]$ as well:

$$\text{SR}^\alpha(q) = \begin{cases} M_S, & \text{if } \overline{S}(q, M_S(q)) \geq \alpha \\ \arg\min_{M_L^{(i)} : S^{(i)}(q, M_S(q)) \leq \overline{S}(q, M_S(q))} C^{(i)}(q), & \text{otherwise.} \end{cases} \tag{11}$$

Here, we use the average score as an aggregate measure of the confidence in adopting the local SLM's output. If the aggregated score is sufficiently high, its answer is adopted directly. Otherwise, we identify the subset of remote LLMs whose corresponding scores fall below the average, indicating that these models are more likely to produce higher-quality answers. Among this subset, we select the one with the lowest expected monetary cost, denoted by $C^{(i)}(q)$, which is computed based on the pricing scheme of the corresponding remote LLM and its average output length (in tokens) observed on the training set.

This rule-based approach for combining multiple scoring model outputs is simple to implement, incurs negligible additional computation, and is intuitively aligned with balancing task performance and monetary cost.

## 4 Experiment

In this section, we present a comparative evaluation of the proposed selective deferred routing against representative existing methods for multi-LLM collaboration on their cost–performance trade-offs. Section 4.1 details the experimental settings, including the LLMs and datasets used, each single LLM's performance, hyperparameters settings, and the baseline methods. Section 4.2 reports performance comparisons between selective deferred routing and the baselines under the single-remote and multi-remote settings, respectively.

### 4.1 Settings

**LLMs.** We employ Qwen-2.5-7B (Qwen Team, 2024) as the local SLM. For remote LLMs, we select Llama-4-Maverick (402B) (Meta AI, 2025), and DeepSeek-V3 (685B) (DeepSeek-AI, 2024) from the family of large-scale open-source models. For proprietary models, we also select GPT-4o (OpenAI, 2024) and o4-mini (OpenAI, 2025) as representatives of general-purpose and reasoning-specialized models, respectively. For the sampling strategy used in acquiring answers, we adopt greedy decoding for the local SLM, while for the remote LLMs, we do not specify any sampling-related parameters to follow the default behavior of the corresponding API providers.

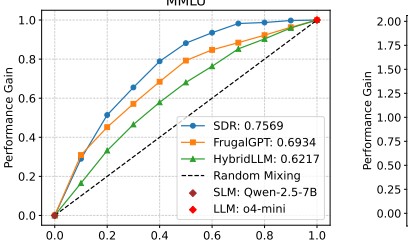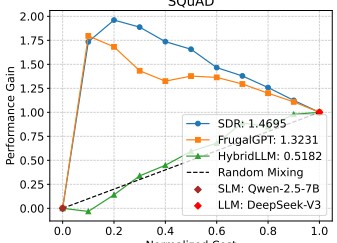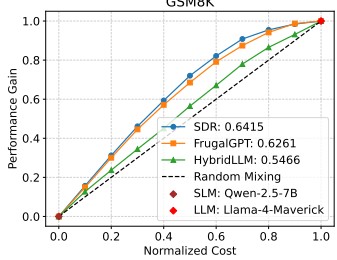

Figure 4: Cost-Performance curves and corresponding AUC obtained from proposed Selective Deferred Routing (SDR) and baseline methods on 3 datasets in Single-Remote Scenario.

**Datasets.** We experiment with a diverse set of datasets covering different aspects of model capabilities: (i) MMLU (Hendrycks et al., 2021): a multiple-choice dataset designed to assess models' broad knowledge and reasoning ability across various domains; (ii) SQuAD (Rajpurkar et al., 2016): a free-form question answering dataset targeting reading comprehension over passages; (iii) GSM8K (Cobbe et al., 2021): a grade-school-level math problem dataset evaluating models' numerical reasoning and problem-solving skills.

Table 1: Local SLM and remote LLMs' average performances in test sets (normalized to 1).

| Model | MMLU | SQuAD | GSM8K |
|---|---|---|---|
| Qwen-2.5-7B | 0.6112 | 0.8188 | 0.3813 |
| Llama-4-Maverick | 0.8893 | 0.8202 | **0.9666** |
| DeepSeek-V3 | 0.8861 | **0.8429** | 0.9644 |
| o4-mini | **0.8911** | 0.7930 | 0.9613 |
| GPT-4o | 0.8747 | 0.8214 | 0.9429 |

Given the varying sizes of these datasets, we adopt different splits for the experiments while ensuring that the same train–test partitions are used across both our proposed method and all baseline methods. Prompts and evaluations are organized consistently across all LLMs, and the resulting performances for each LLM on datasets are shown in Table 1. More detailed information on dataset usage is provided in Appendix B.

**Hyperparameters.** For the threshold $\beta$ in Eq. 8, which determines whether the remote LLM offers a significant advantage, we report results with $\beta = 0$ in the main text, while deferring results under alternative values to Appendix C.2. For the loss function defined in Eq. 9, we follow the recommendation of the original paper (Yuan et al., 2021) by initializing $a$ and $b$ to 0 and setting $m = 1$.

**Baselines.** We compare our proposed method against several representative multi-LLM collaboration methods: (i) FrugalGPT (Chen et al., 2023): it evaluates SLM responses with a trained DistillBERT (Sanh et al., 2019) model and routes low-quality queries to the LLM; (ii) HybridLLM (Ding et al., 2024): it employs a trained DeBERTa (He et al., 2020) model to estimate the relative quality of SLM vs. LLM outputs and selects accordingly with cost taken into account; (iii) RouterBench (Hu et al., 2024): it encodes queries using Sentence Transformer (Reimers & Gurevych, 2019) and later predicts all available LLMs' response quality with KNN and MLP models, followed by cost-aware selection. Given their respective design scopes, we compare against (i) and (ii) in the single-remote setting, and against (iii) in the multi-remote setting.

## 4.2 RESULTS

**Single-Remote Scenario.** In this setting, we select the best-performing LLM on each dataset (highlighted in bold in Table 1) as the remote LLM. The comparative results between the proposed Selective Deferred Routing (SDR) and the baseline methods are presented in Figure 4. We report both the Cost-Performance Curve defined in Section 3.2 and the corresponding AUC metric (annotated in the legend of each method). For readability of the plots, we sample data points by choosing trade-off parameters, which are varies across methods, such that the normalized cost defined in Eq. 3 is approximately uniformly distributed over $[0, 1]$. The AUC metric provides a more precise measure, as it is computed analytically by considering over all possible trade-off parameter values[1].

---

[1] Readers interested in the detailed derivation may refer to Appendix A

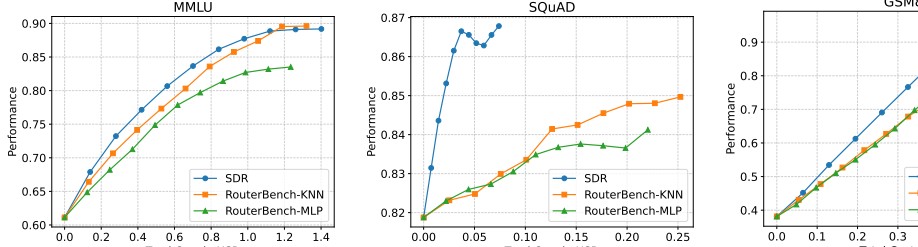

Figure 5: Cost-Performance curves of real monetary costs (in USD) and original performance obtained from proposed Selective Deferred Routing (SDR) and baseline methods on 3 datasets in Multi-Remote Scenario.

As shown in the figure, Selective Deferred Routing consistently outperforms baseline methods across almost all data points and achieves the highest AUC on every dataset. This indicates that SDR can deliver strong performance for users with different preferences on the cost–performance trade-off, demonstrating the effectiveness of its optimization objective and scoring model design.

It is worth noting that on the SQuAD dataset, the SDR method achieves its optimal performance with a normalized cost of about 0.2 (i.e., fewer than a quarter of the queries are routed to the remote LLM). Remarkably, this even surpasses the performance of the remote LLM by a large margin. One of the reasons is that when the performance gap between the local SLM and the remote LLM on a given task is not significant, there will be queries for which the local SLM performs better. SDR can effectively identify some of those queries, thereby leveraging selective routing to attain an overall performance superior to that of any single LLM.

We also find that on the GSM8K dataset, the performance gap between SDR and FrugalGPT is the smallest, since the remote LLM achieves nearly perfect accuracy on this dataset (as shown in Table 1). In this case, the performance gap between the SLM and the LLM is almost entirely determined by whether the SLM's answer is correct, which coincides with the straightforward training objective adopted by FrugalGPT.

**Multi-Remote Scenario.** In this setting, we involve all LLMs mentioned in Section 4.1. The comparative results between the proposed Selective Deferred Routing (SDR) and the baseline methods are presented in Figure 5. Since the cost-performance AUC defined in Section 3.2 for the single-remote scenario is no longer applicable, we report the curves formed by the actual monetary cost and original performance achieved by different methods. The monetary cost is computed based on the pricing schemes of different LLMs (detailed in Appendix B) and the number of input and output tokens. Data points are selected as follows: we first vary each method's trade-off parameter to identify the points yielding the lowest cost and the highest performance, thereby determining the overall trade-off range. We then sample intermediate points at a specified rate (e.g., 10 intervals in the figure), ensuring that the resulting costs are approximately evenly distributed within this range.

As illustrated in the figure, across all datasets considered, Selective Deferred Routing consistently surpasses the baseline methods at nearly every sampled point, indicating that it delivers strong effectiveness under diverse preferences in the cost–performance trade-off. These results demonstrate that our rule-based extension from the single-remote to the multi-remote setting remains both lightweight and effective.

## 5 CONCLUSION

In this paper, we propose Selective Deferred Routing (SDR), a paradigm for cost-efficient collaboration between local SLMs and remote LLMs. SDR first leverages the local SLM to generate a preliminary answer along with other informative outputs, which are subsequently exploited for selective routing decisions. A trained decider module evaluates the preliminary answer's quality, and selectively routes them to the most suitable remote LLM to obtain higher-quality responses. Extensive experiments conducted on five LLMs across three datasets demonstrate that the proposed method consistently outperforms representative baseline methods under different trade-off preferences in both single-remote and multi-remote scenarios.

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

## A  Proof of the Optimal BSR Theorem

According to the definition of Cost in Eq. 3, for a finite dataset $\mathcal{Q}$, the possible values of Cost are restricted to $\{\frac{i}{|\mathcal{Q}|} \mid i \in [0, |\mathcal{Q}|]\}$.

For all $q \in \mathcal{Q}$, we sort the elements in ascending order according to their scores given by the scoring model, $S(q)^2$, resulting in the sequence $(q_{(1)}, q_{(2)}, \cdots, q_{(|\mathcal{Q}|)})$. We can then establish a one-to-one correspondence between the range of the decision threshold $\alpha$ and the resulting Cost:

$$
\begin{cases}
\alpha \in (-\infty, S(q_{(1)})] \Leftrightarrow \text{Cost}(\mathcal{Q}, \text{BSR}^\alpha) = 0, \\
\alpha \in (S(q_{(i)}), S(q_{(i+1)})] \Leftrightarrow \text{Cost}(\mathcal{Q}, \text{BSR}^\alpha) = \frac{i}{|\mathcal{Q}|}, \forall i \in [1, |\mathcal{Q}|), \\
\alpha \in (S(q_{(|\mathcal{Q}|)}), \infty) \Leftrightarrow \text{Cost}(\mathcal{Q}, \text{BSR}^\alpha) = 1.
\end{cases}
$$

Therefore, we can select a representative $\alpha$ from each except the first[3] of the above intervals to form a sequence $(\alpha_1, \cdots, \alpha_{|Q|})$ to reformulate the expected performance gain in Eq. 6 as follows:

$$
\mathbb{E}[\text{PG}] = \frac{1}{|\mathcal{Q}|} \sum_{i=1}^{|\mathcal{Q}|} \text{PG}(\mathcal{Q}, \text{BSR}^{\alpha_i}).
$$

According to the definitions in Eq. 5 and Eq. 7, the performance gain can be computed as the prefix sum of the performance gaps along the sequence of $q$ sorted as described above:

$$
\text{PG}(\mathcal{Q}, \text{BSR}^{\alpha_i}) = \sum_{j=1}^{i} \text{Gap}(q_{(j)}).
$$

Consequently, the expected performance gain can be further rewritten as:

$$
\mathbb{E}[\text{PG}] = \frac{1}{|\mathcal{Q}|} \sum_{i=1}^{|\mathcal{Q}|} \sum_{j=1}^{i} \text{Gap}(q_{(j)}).
$$

From the form of the above expression, it follows directly from the rearrangement inequality that the sum of prefix sums is maximized when the array is sorted in non-increasing order. Hence, Theorem 1 is established.

## B  Additional Details of the Experiment Settings

**Datasets Using.** We use the test split of the MMLU (Hendrycks et al., 2021) and the validation splits of the SQuAD (Rajpurkar et al., 2016), further dividing each into training and test sets with an 8:2 ratio, since their original training splits are excessively large. For GSM8K (Cobbe et al., 2021), we directly adopt its official training and test splits.

For prompt organization, we adapt to the nature of each dataset: in MMLU and GSM8K, we allow the LLMs to generate intermediate reasoning steps, with the final answer explicitly formatted at the end of the response to facilitate answer retrieval for evaluation. For SQuAD, we require the LLMs to output answers directly.

For evaluation, correctness on MMLU and GSM8K is determined by exact matching with the reference answers. For SQuAD, we follow the official recommendation and use the maximum F1 score between the model's output and the multiple reference answers as the measure of answer quality.

**LLMs Pricing.** We assume that the local SLM does not incur any monetary cost. The pricing data of the remote LLMs used in our experiments are summarized in Table 2. For the open-source models Llama-4-Maverick (Meta AI, 2025), and DeepSeek-V3 (DeepSeek-AI, 2024), we adopt the average pricing across all API providers listed on Artificial Analysis. For the proprietary models (o4-mini and GPT-4o), we follow their official API pricing (OpenAI).

---

[2]In this section, we abbreviate $S(q, M_S(q))$ as $S(q)$.

[3]We do not need to select an $\alpha$ from the interval corresponding to $\text{Cost}(\mathcal{Q}, \text{BSR}^\alpha) = 0$, as this would not yield any performance gain.

Table 2: LLMs pricing data used in experiments.

| Model | Input (USD per million Tokens) | Output (USD per million Tokens) |
|---|---|---|
| Llama-4-Maverick | 0.29 | 0.93 |
| DeepSeek-V3 | 1.00 | 1.54 |
| o4-mini | 1.10 | 4.40 |
| GPT-4o | 2.50 | 10.00 |

## C  SUPPLEMENTARY EXPERIMENTS

### C.1  SCORING MODEL WITH DIFFERENT NUMBERS OF TRANSFORMER LAYERS

We also implemented and evaluated scoring models composed of multiple transformer layers. Similar to the single-layer design described in the main text, suppose the local SLM consists of $n$ transformer layers and the scoring model includes $m$ layers. In this case, the $m$ layers of the scoring model are initialized using the pretrained parameters of the $(n - m + 1)$-th through $n$-th layers of the local SLM, while the hidden states output from the $(n - m)$-th layer of the local SLM are used as input.

Table 3: AUC obtained from scoring models with different numbers of transformer layers.

| Layers | MMLU | SQuAD | GSM8K |
|---|---|---|---|
| $m = 1$ | 0.7569 | 0.7585 | 0.7555 |
| $m = 2$ | 1.4695 | 1.4416 | 1.5249 |
| $m = 3$ | 0.6415 | 0.6423 | 0.6435 |

The experimental results are summarized in Table 3. As shown, increasing the number of layers in the scoring model does not yield a consistent improvement in the AUC metric across all datasets. Consequently, we adopt the simplest and most lightweight single-layer design as our final choice.

### C.2  DIFFERENT VALUES OF BINARY GAP THRESHOLD

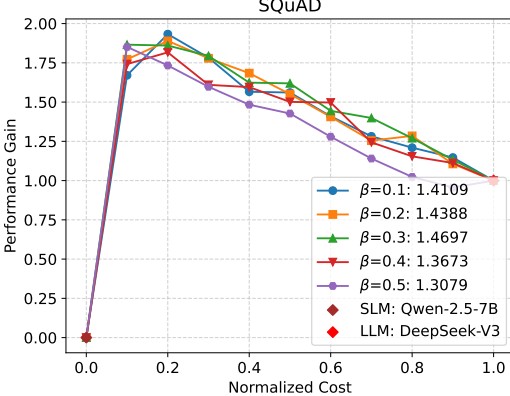

Figure 6: Cost-Performance curves and corresponding AUC obtained from proposed Selective Deferred Routing (SDR) with 5 different $\beta$ on SQuAD in Single-Remote Scenario.

The parameter $\beta$ represents the tolerance toward local SLM answers: only when the remote LLM's answer exceeds the local one by more than $\beta$ is it regarded as having a significant advantage. Accordingly, its value lies within $[0, 1)$. For the MMLU and GSM8K datasets, the performance gap can

only take values from $\{-1, 0, 1\}$, making different settings of $\beta$ effectively equivalent. On SQuAD, the results under five different $\beta$ values are shown in Figure 6. We observe that when $\beta$ is relatively small (0.1–0.3), the increased tolerance toward local SLM answers allows the scoring model to achieve better performance under more limited budgets. In contrast, when $\beta$ is larger (above 0.3), the scoring model's effectiveness drops notably, possibly because the binary dataset determined by $\beta$ becomes imbalanced, thereby degrading training effectiveness.

## D  USE OF LARGE LANGUAGE MODELS

During the preparation of this manuscript, large language models (LLMs) were employed solely for language polishing, with the aim of enhancing clarity and readability without altering the technical content or the authors' original intent.

