# OpenReview forum: "Selective Deferred Routing: Enabling Cost-Efficient Collaboration between Local SLMs and Remote LLMs"
_ICLR.cc/2026/Conference — Submitted to ICLR 2026_

### Official Review · Reviewer_iSCG · 2025-10-31

**Soundness:** 2
**Presentation:** 3
**Contribution:** 2
**Rating:** 4
**Confidence:** 4

**Summary:**

The paper introduces Selective Deferred Routing (SDR), a cost-efficient paradigm that optimizes collaboration between local small language models (SLMs) and remote large language models (LLMs). The approach leverages local models to generate preliminary responses and utilizes a decider module to route requests to the most suitable remote LLM for enhanced output. The paper demonstrates that SDR consistently outperforms existing multi-LLM collaboration methods by improving the cost–performance trade-offs across a variety of tasks and datasets, with extensive experimental evaluations involving five LLMs and three datasets.

**Strengths:**

1. The idea of combining local SLMs and remote LLMs in a selective and cost-efficient manner is novel. It addresses the challenge of balancing monetary cost and task performance, a significant concern in practical applications of LLMs.

2. The proposed method is well-structured, with clear descriptions of the decider module and the associated scoring model, backed by a strong theoretical foundation. The experiments demonstrate the effectiveness of SDR, showing consistent improvements over baseline methods in the single-remote and multi-remote scenarios.

3. The paper is generally well-written, with clear explanations of the methodology, theoretical formulations, and experimental setups. The figures, such as the cost-performance curves and AUC metrics, effectively illustrate the improvements achieved by SDR

4. The approach can significantly reduce the operational costs of LLMs by optimizing the trade-off between local and remote model performance. The flexibility of the method for various user preferences makes it highly applicable in real-world systems that deploy LLMs.

**Weaknesses:**

1. While SDR offers an interesting method for cost optimization, it shares conceptual similarities with existing research, particularly Immediate Routing and Model Cascades. The paper could have further highlighted how SDR uniquely overcomes the limitations of these existing methods, such as the sequential inefficiency in Model Cascades or the oversimplification of Immediate Routing.

2. The experimental settings section (pages 8–9) provides a good overview of model configurations, but the hyperparameter tuning details, especially for the decider module and thresholds, are vague. Without a detailed explanation of how these parameters were optimized, it's unclear how much the performance gains are attributable to fine-tuning versus the intrinsic qualities of SDR. Also, it would be beneficial to provide open-source code to ensure full reproducibility.

3. Although SDR performs well in multi-remote scenarios, the scalability of the model with multiple remote LLMs might pose practical issues, particularly regarding memory and computation. The parallel running of multiple scoring models during deployment, though lightweight, may still lead to increased computational overhead in large-scale systems.

**Questions:**

1. The paper discusses the use of a threshold parameter (α) for routing decisions (Equations 2 and 3). Can you elaborate more on how these thresholds are dynamically adjusted in real-world use cases, and whether the model adapts to changing task characteristics over time?

2. The experiments focus on single-task scenarios, but many real-world applications of LLMs require multi-task performance. How well does the Selective Deferred Routing approach perform when tasks with vastly different characteristics (e.g., creative writing vs. technical problem-solving) are combined?

3. The paper mentions cost-effective collaboration, but what happens in scenarios with limited cloud resources or heavy latency on the edge devices? Is SDR robust enough to handle extremely resource-constrained environments, and how does it compare to other edge-cloud hybrid models?

4.  Given the lightweight nature of the decider module, what are the potential latency concerns when scaling SDR to a production environment with high-frequency queries? Are there any optimizations or architectural changes that can further reduce response time in such scenarios?

---

> ### Author Response · Authors · 2025-11-28
>
> Dear Reviewer iSCG,
>
> We deeply appreciate the time and care you have invested in reviewing our paper and for the insightful comments you kindly provided. Below, we offer detailed responses to the concerns and questions you raised.
>
> ## Differences between the proposed SDR and existing methods
> As elaborated in Section 1, SDR differs from prior approaches in two key ways. First, SDR parallelizes the selection of remote LLMs, thereby avoiding the high latency and application constraints inherent to model cascades, which rely on a linear structure. Second, SDR mitigates the information scarcity issue associated with immediate routing by better leveraging the capabilities of the local SLM.
>
> ## Hyperparameter settings
> The hyperparameters used in our implementation have been specified in Section 4.1, and Appendix C further explores the sensitivity of the algorithm to additional hyperparameter choices.
>
> ## Potential scalability issues arising from running multiple scoring models in parallel
> As noted in Section 3.3, the scoring model contains an order of magnitude fewer parameters than the SLM itself. This implies that even if one considers more than 10 remote LLM candidates, the local device’s memory consumption would not double. We believe this is sufficient for realistic deployment scenarios, as the number of truly top-performing LLMs available on the market, despite ongoing iteration, remains limited. It is unlikely that hundreds of distinct, mutually competitive frontier LLMs would exist simultaneously.
>
> ## The tuning of $\alpha$
> Sections 3.1–3.2 clarify that users who prioritize lower cost should set a smaller $\alpha$, while users who prefer higher performance should increase accordingly.
>
> ## Task diversity
> The datasets we selected cover as broad a range of mainstream LLM applications as possible, including fundamental knowledge tasks, mathematical reasoning, and reading comprehension. These results demonstrate that the proposed method maintains strong effectiveness across several widely used application domains.
>
> ## Resource-limited Scenarios
> We believe this topic, while highly relevant to real-world deployments, falls outside the primary scope of this work. Nonetheless, we offer a brief comparison with edge–cloud hybrid models based on your suggestion.
> Hybrid models dynamically deploy a single model across both edge and cloud, requiring each request to run on both sides. Their performance is therefore consistently affected by cloud-side load and edge–cloud communication efficiency. In contrast, under SDR, requests that the local SLM is capable of handling remain entirely on the local device, reducing dependency on cloud workload or edge-cloud communication.
>
> ## High-frequency scenarios
> The primary use case we envision for SDR is personal users or small organizations seeking to reduce remote API cost by effectively utilizing lower-performance local devices. Such users are unlikely to encounter high-frequency request patterns. Nonetheless, extending this paradigm to fully cloud-based, high-throughput deployments could be a promising direction for future exploration, and we appreciate your insightful comment.
>
> Once again, we sincerely thank you for your careful review and valuable feedback. We hope that our responses adequately address all of your concerns.
>
> Sincerely,
> All authors

---

### Official Review · Reviewer_kLrF · 2025-11-02

**Soundness:** 2
**Presentation:** 2
**Contribution:** 2
**Rating:** 2
**Confidence:** 2

**Summary:**

The paper introduces Selective Deferred Routing (SDR), a two-stage collaboration scheme: a local SLM first answers and emits hidden-state features; a lightweight “decider” then either accepts the local answer or routes once to a remote LLM, aiming to optimize a cost–performance tradeoff. The core contribution is to cast binary selective routing as an AUC-style objective over a cost–performance curve, leading to a ranking-consistent training loss (via a Binary-Gap surrogate) for the decider initialized from a single SLM transformer layer. Experiments over 3 datasets and 5 LLMs show favorable tradeoffs against routing/cascade baselines, plus a simple rule to extend to multi-remote settings.

**Strengths:**

- Clear formalization of selective routing with an AUC objective that directly targets cost–performance trade-offs; the gap-ordering optimality condition is crisp and intuitive.

- Lightweight decider design that reuses a single Transformer layer from the local SLM; practical and latency-friendly.

- Empirical results cover five LLMs / three datasets, reporting normalized AUC curves (single-remote) and actual USD costs (multi-remote).

**Weaknesses:**

- Latency measurements and on-device memory/compute overheads for running the decider in parallel with the SLM are discussed qualitatively, but not benchmarked.

- Label generation cost is under-specified: training the decider requires evaluating both local and remote outputs per query to estimate gaps/BG labels; the offline token cost could rival the savings at deployment time, but is not quantified.

- Baselines omit some recent routers/cascades in the multi-LLM literature beyond those cited; fairness of API defaults (temperatures, system prompts) across providers is not detailed.

- Generalization across local models is unclear: the decider is initialized from a specific SLM layer; portability to different SLMs or quantization variants is not evaluated.

**Questions:**

- What is the total offline cost (tokens × price) to collect BG labels per dataset and per remote LLM, and after how many live queries does SDR break even?

- How sensitive is SDR to changing the local SLM (e.g., different size/architecture, quantization levels)? Can the decider trained on one SLM transfer without re-labeling?

- For the multi-remote case, why is averaging the scores the right aggregation? Have you compared to learned policies (e.g., budget-constrained bandits or cost-aware ranking) or provided a regret bound?

---

> ### Author Response · Authors · 2025-11-28
>
> Dear Reviewer kLrF,
>
> We sincerely thank you for the time and effort you have devoted to reviewing our manuscript and for offering thoughtful and constructive comments. Below, we provide detailed responses to your concerns and questions.
>
> ## The deployment performance of the proposed method
> >Latency measurements and on-device memory/compute overheads for running the decider in parallel with the SLM are discussed qualitatively, but not benchmarked.
>
> The SDR method introduced in this work is primarily designed to optimize the trade-off between monetary cost and task performance. The lightweight design is intended to ensure minimal additional deployment overhead and thereby preserve the practical applicability of the method. It is **not specifically aimed at optimizing for the deployment side**, which is why the current version of the manuscript does not include furthur analyses or experiments.
>
> That said, we acknowledge that implementing a system prototype and incorporating in-practice evaluations in future revisions would further strengthen the practical value of our approach, and we thank you for pointing out this aspect.
>
> ## The label generation cost
> >Label generation cost is under-specified......and after how many live queries does SDR break even?
>
> First, the evaluation process adopted in our work, to the best of our knowledge, basically consistent with other works in this research area, focuses primarily on assessing the effectiveness of the training method. The cost of generating labels is **determined by the size of the dataset rather than by the method design**, and thus has limited relevance to comparing the effectiveness of different approaches.
>
> As for how long it would take for online deployment to amortize the training cost, to the best of our knowledge, all relevant methods in this domain require collecting LLM responses over the dataset (though the specific form may vary). Consequently, the better the cost–performance trade-off a method achieves, the sooner it can offset the training overhead.  A brief estimation is as follows. On the MMLU dataset (Figure 4(1)), if we set the performance target to 90% of the LLM’s accuracy, **the baselines need to route approximately 80% of the queries to the LLM, whereas SDR requires less, about 60%**. Assuming that SDR and the baselines are trained on datasets of the same size (that's what we exactly do in the experiment), and that the cost of processing each query with the LLM is identical, then to compensate for the label generation cost, the baselines would need to online process **5 times the size of the training dataset** $(1 / (1 – 80\%))$, while SDR would require only 2.5 times $(1 / (1 – 60\%))$. This corresponds to roughly a two-fold improvement from that point of view.
>
> Furthermore, label generation cost can be significantly reduced with contributions from the open-source community. For example, RouterBench has released responses of 11 mainstream LLMs across 8 datasets, substantially lowering the training cost for subsequent work.
>
> ## More baselines and literature review
>
> We selected for comparison several of the most classical and representative works in the Multi-LLM routing literature. In Section 2 (Related Work), we also include brief descriptions of other more up-to-date methods to the best of our knowledge. However, due to limited resources, it is not feasible for us to reproduce and compare against every existing approach. If there are specific works among those we might have overlooked that you believe are particularly significant and valuable for comparison, we would greatly appreciate your pointing them out. We will be happy to include them in future revisions.
>
> ## The specific parameters used for remote APIs and the exact prompt formatting
>
> As stated in the paper, we ensured **consistent configurations when evaluating both our method and the baselines**, which is sufficient for a fair comparison of their effectiveness. But including these details in the appendix would indeed enhance the completeness of the paper. We will incorporate those information in a future revision, and we appreciate your valuable suggestion.
>
> (continued in next comment)

---

> > ### Author Response · Authors · 2025-11-28
> >
> > (cont’d from previous comment)
> >
> > ## Generalization to different SLMs
> >
> > >How sensitive is SDR to changing the local SLM?
> >
> > During the rebuttal period, we conducted additional experiments in the one-remote scenario using an SLM from a different model family (Llama3.1-8B), while keeping all other experimental settings identical to those described in Section 4.1 of the paper. The following table reports the AUC comparison between our SDR method and the baseline methods. As shown, the effectiveness of our approach extends to a completely different model architecture.
> >
> > |Method \ Dataset | MMLU | SQuAD | GSM8K |
> > | --- | --- | --- | --- |
> > |FrugalGPT | 0.6361   | 0.8493   | 0.5054   |
> > |HybridLLM | 0.5371   | 0.5404   | 0.5031   |
> > |SDR (ours)| **0.6714**  | **0.8557**   | **0.5062**    |
> >
> > Moreover, as you suggested, we will continue to include experiments with SLMs of different parameter scales and quantization configurations in future revisions. However, we would like to note that the variations introduced by different parameter sizes or quantization levels within the same model family are typically much smaller than the architectural differences across model families. Therefore, the above results already provide strong evidence of the generalizability of our method across different SLMs.
> >
> > >Can the decider trained on one SLM transfer without re-labeling?
> >
> > Labels essentially reflects the relative capability between the SLM and the LLM on a given dataset. If we replace the SLM, it is unavoidable that we must collect the new SLM’s responses on the dataset, since different models naturally exhibit non-negligible performance differences. On the other hand, if no new remote LLM is introduced, the LLM response labels can be fully reused.
> >
> > ## Exploring additional approaches in the multi-remote scenario
> > >For the multi-remote case, why is averaging the scores the right aggregation? Have you compared to learned policies (e.g., budget-constrained bandits or cost-aware ranking) or provided a regret bound?
> >
> > The directions you suggested are indeed promising. However, the primary focus of this work is to introduce improved training objectives and module designs within a single-remote setting. And a simple aggregation strategy already produces performance superior to the baselines in the multi-remote scenario is an encouraging byproduct, demonstrating preliminary extensibility across different application settings. Exploring more sophisticated decision mechanisms under multi-remote configurations would undoubtedly further unlock the potential of our method and represents an important direction for future research. We appreciate your constructive suggestion.
> >
> > Once again, we are grateful for your thoughtful review and valuable feedback. We hope that our responses have satisfactorily addressed all of your concerns.
> >
> > Sincerely,
> > All authors

---

### Official Review · Reviewer_cQJE · 2025-11-03

**Soundness:** 2
**Presentation:** 2
**Contribution:** 2
**Rating:** 4
**Confidence:** 3

**Summary:**

This paper proposes Selective Deferred Routing, a paradigm that enables cost-efficient collaboration between local SLMs and remote LLMs. In this framework, a user request is first processed by the local SLM, which not only generates a preliminary response but also provides rich semantic representations of the request. A lightweight decider module then leverages this information to either adopt the initial response or route the request in a single step to the most suitable remote LLM for a higher-quality response.

**Strengths:**

1. Multi-LLM Routing is important and on time.

2. Selective Deferred Routing balances cost and accuracy.

3. Extensive experiments on 5 LLMs and 3 datasets are provided.

**Weaknesses:**

1. The latency is not reported. Judge after SLM finishes generation can take very long. After SLM generates a few tokens, it might be sufficient to start routing.

2. Fine tuning a BERT before SLM and LLMs may have lower latency and cost.

3. In GSM8K Figure 4, the performance is close to FrugalGPT. Code is also not provided.

**Questions:**

In GSM8K Figure 4, why using Llama-4-Maverick as the LLM? How about GPT and Deepseek?

---

> ### Author Response · Authors · 2025-11-28
>
> Dear Reviewer cQJE,
>
> We sincerely appreciate the time and effort you devoted to reviewing our manuscript and for providing insightful comments and suggestions. Below, we provide detailed responses to your concerns and questions.
>
> ## The latency of the proposed method
> >The latency is not reported.
>
> The SDR method introduced in this work is primarily designed to optimize the trade-off between monetary cost and task performance. The lightweight design is intended to ensure minimal additional deployment overhead and thereby preserve the practical applicability of the method. It is **not specifically aimed at optimizing latency**, which is why the current version of the manuscript does not include latency-focused analyses or experiments.
>
> That said, we acknowledge that incorporating latency evaluations in future revisions would further strengthen the practical value of our approach, and we thank you for pointing out this aspect.
>
> ## The idea of performing routing earlier
> >Judge after SLM finishes generation can take very long. After SLM generates a few tokens, it might be sufficient to start routing.
>
> We agree that this is a valuable idea, but it introduces new challenges and may drift away from the core research question of the cost–performance trade-off. A typical example can help illustrate this: current LLMs often exhibit extensive **reflective–refining** behavior during reasoning. And you have likely observed cases where they repeatedly output phrases such as "Wait, let’s see......" before revising their answers. If routing decisions were made immediately after the SLM produces an initial, unrefined draft, the reasonable behavior would likely be routing to the remote LLM prematurely, even in cases where the local SLM might eventually reach a correct answer on its own.
>
> Consequently, exploring this direction would likely lead to a new trade-off between latency and performance and possibly require additional designs regarding to SLM behavior. Overall, while this line of inquiry deviates from our original goal of optimizing the cost–performance trade-off, it indeed represents a valuable direction for future extension. We sincerely appreciate you sharing this insightful thought with us.
>
> ## Latency & cost versus finetuning a BERT model
> >Fine tuning a BERT before SLM and LLMs may have lower latency and cost.
>
> As described in Section 3.3, the decider module in our method is constructed primarily by reusing one transformer layer from the SLM. With a 7B-scale SLM, the additional module contains approximately 200M parameters, **comparable to the scale of BERT**. Moreover, the module can operate partially in parallel with the original SLM, so qualitatively it will not introduce greater latency than a BERT-based approach.
>
> Concerning training cost, a straightforward implementation indeed requires running the original SLM online to obtain hidden states, which requires memory to store the SLM parameters. However, since our method does not update the SLM parameters, no additional memory is needed for activations or similar information of SLM. Furthermore, if sufficient disk storage is available, one can precompute and store the hidden states offline and simply load them during training, thus avoiding online SLM inference entirely.
>
> ## The experimental results on GSM8K
> The relatively small advantage over FrugalGPT on GSM8K is explained in Section 4.2 of the manuscript. This is due to the near-saturated performance of LLMs on this dataset, under which the **new training objective we propose becomes intrinsically similar to those used in prior work**. The reason for using Llama-4 as the LLM for this dataset is also stated in the first sentence of Section 4.2: this model achieves the strongest overall performance on GSM8K. This decision was made due to space limits; however, we acknowledge that adding the experimental results for all remaining LLMs to the appendix would enhance the completeness of the paper.
>
> Finally, we once again thank you for your thoughtful review and valuable suggestions. We hope that our responses have satisfactorily addressed all of your concerns.
>
> Sincerely,
> All authors

---

### Meta-Review · Area_Chair_heCt · 2026-01-06

**Summary:**

The reviewers find the paper to be timely and clearly written, proposing Selective Deferred Routing as a framework for balancing monetary cost and performance by coordinating local SLMs with remote LLMs. While the formulation of routing as a cost–performance optimization problem and the AUC-style training objective are viewed as reasonable, reviewers consistently raised concerns about the practical and empirical grounding of the approach. In particular, the lack of latency measurements, insufficient accounting of offline label generation cost, limited discussion of deployment overhead, and unclear advantages over closely related routing and cascade methods reduce confidence in the paper’s overall impact and readiness

**Reviewer Concerns:**

Although the rebuttal provided clarifications on design choices, additional experiments on a different SLM, and qualitative arguments about deployment and training cost, key concerns regarding missing latency benchmarks, unquantified offline labeling cost, incomplete baseline coverage, and limited evidence of practical scalability and deployment efficiency remain largely unresolved.

**Reviewer Scores:**

Reviewers with borderline scores would likely maintain their original assessments after discussion, while the reviewer recommending rejection would keep the same score, resulting in an overall consensus that the paper does not yet meet the bar for acceptance.

---

### Decision · Program_Chairs · 2026-01-26

Reject